# African Genomic Medicine Portal: A Web Portal for Biomedical Applications

**DOI:** 10.3390/jpm12020265

**Published:** 2022-02-11

**Authors:** Houcemeddine Othman, Lyndon Zass, Jorge E. B. da Rocha, Fouzia Radouani, Chaimae Samtal, Ichrak Benamri, Judit Kumuthini, Yasmina J. Fakim, Yosr Hamdi, Nessrine Mezzi, Maroua Boujemaa, Chiamaka Jessica Okeke, Maureen B. Tendwa, Kholoud Sanak, Melek Chaouch, Sumir Panji, Rym Kefi, Reem M. Sallam, Anisah W. Ghoorah, Lilia Romdhane, Anmol Kiran, Ayton P. Meintjes, Perceval Maturure, Haifa Jmel, Ayoub Ksouri, Maryame Azzouzi, Mohammed A. Farahat, Samah Ahmed, Rania Sibira, Michael E. E. Turkson, Alfred Ssekagiri, Ziyaad Parker, Faisal M. Fadlelmola, Kais Ghedira, Nicola Mulder, Samar Kamal Kassim

**Affiliations:** 1Sydney Brenner Institute for Molecular Bioscience, Faculty of Health Sciences, University of the Witwatersrand, 9 jubilee Road, Parktown, Johannesburg 2193, South Africa; houcemeddine.othman@wits.ac.za (H.O.); jdarocha1@gmail.com (J.E.B.d.R.); 2Computational Biology Division, Department of Integrative Biomedical Sciences, IDM, CIDRI Africa Wellcome Trust Centre, University of Cape Town, Rondebosch, Cape Town 7701, South Africa; lyndon.zass@uct.ac.za (L.Z.); sumir.panji@uct.ac.za (S.P.); aytonm@gmail.com (A.P.M.); perceval.maturure@uct.ac.za (P.M.); ziyaad.parker@uct.ac.za (Z.P.); nicola.mulder@uct.ac.za (N.M.); 3Division of Human Genetics, National Health Laboratory Service, School of Pathology, Faculty of Health Sciences, University of the Witwatersrand, Johannesburg 2001, South Africa; 4Chlamydiae and Mycoplasma Laboratory, Research Department, Institut Pasteur du Maroc, Casablanca 20360, Morocco; radouani@gmail.com (F.R.); ichrakbenamri@gmail.com (I.B.); kholoud.sanak@gmail.com (K.S.); maryame.azz@gmail.com (M.A.); 5Laboratory of Biotechnology, Environment, Agri-Food and Health, Faculty of Sciences Dhar El Mahraz, Sidi Mohammed Ben Abdellah University, Fez 30000, Morocco; samtalchaimae@gmail.com; 6Systems and Data Engineering Team, National School of Applied Sciences, Abdelmalek Essaâdi University, Tangier 93000, Morocco; 7South African Bioinformatics Institute, University of Western Cape, Cape Town 7535, South Africa; jkumuthini@gmail.com; 8Biotechnology Unit, Department of Agricultural and Food Science, Faculty of Agriculture, University of Mauritius, Reduit 80837, Mauritius; yasmina@uom.ac.mu; 9Laboratory of Biomedical Genomics and Oncogenetics, LR20IPT05, Institut Pasteur de Tunis, University of Tunis El Manar, Tunis 1002, Tunisia; yosr.hamdi@pasteur.utm.tn (Y.H.); mezzinessrine1@gmail.com (N.M.); maroua.boujemaa@gmail.com (M.B.); rym.kefi@pasteur.tn (R.K.); lilia.romdhane@pasteur.utm.tn (L.R.); jmelhaifa@gmail.com (H.J.); 10Laboratory of Human and Experimental Pathology, Institut Pasteur de Tunis, Tunis 1002, Tunisia; 11Department of Biology, Faculty of Science of Bizerte, Jarzouna, Bizerte 7021, Tunisia; 12Research Unit in Bioinformatics (RUBi), Department of Biochemistry and Microbiology, Rhodes University, Makhanda 6139, South Africa; okekechiamaka39@gmail.com (C.J.O.); bilingatendwa@gmail.com (M.B.T.); 13Laboratory of Bioinformatics, Biomathematics and Biostatistics, Pasteur Institute of Tunisia, University of Tunis El Manar, Tunis 1068, Tunisia; mcmelek@msn.com (M.C.); kais.ghedira@pasteur.tn (K.G.); 14Basic Medical Science Department, Faculty of Medicine, Galala University, Suez 43727, Egypt; reemsallam@med.asu.edu.eg; 15Medical Biochemistry and Molecular Biology Department, Faculty of Medicine, Ain Shams University, Cairo 11381, Egypt; 16Department of Digital Technologies, Faculty of Information, Communication and Digital Technologies, University of Mauritius, Reduit 80837, Mauritius; a.ghoorah@uom.ac.mu; 17Malawi-Liverpool-Wellcome Trust, Queen Elizabeth Central Hospital, College of Medicine, Blantyre P.O. Box 30096, Malawi; anmol.kiran@gmail.com; 18Department of Clinical Infection, Microbiology and Immunology, Institute of Infection and Global Health, University of Liverpool, Liverpool L69 3BX, UK; 19Laboratory of Venoms and Therapeutic Molecules (LR16IPT08), 1" Place Pasteur, BP74, Pasteur Institute of Tunis, University of Tunis El Manar, Tunis 1002, Tunisia; ayoub.ksouri@pasteur.utm.tn; 20Compute Science Department, The Higher Future Institute for Specialized Technological Studies, Cairo 11381, Egypt; Mohammed_Farahat19@yahoo.com; 21Centre for Bioinformatics and Systems Biology, Faculty of Science, University of Khartoum, Khartoum 11115, Sudan; samah.ahmed8848@gmail.com (S.A.); ransibeirah3@gmail.com (R.S.); faisalfadl@gmail.com (F.M.F.); 22National Institute for Mathematical Sciences, PMB Kwame Nkrumah University of Science and Technology (KNUST), Kumasi, Ghana; eboturkson@nims.edu.gh; 23Division of Entomology and Core Molecular Biology, Bioinformatics Facility, Uganda Virus Research Institute, Entebbe P.O. Box 49, Uganda; assekagiri@gmail.com; 24MASRI Research Institute, Ain Shams University, Cairo 11381, Egypt

**Keywords:** Africa, clinical genetics, database, genomic medicine, portal

## Abstract

Genomics data are currently being produced at unprecedented rates, resulting in increased knowledge discovery and submission to public data repositories. Despite these advances, genomic information on African-ancestry populations remains significantly low compared with European- and Asian-ancestry populations. This information is typically segmented across several different biomedical data repositories, which often lack sufficient fine-grained structure and annotation to account for the diversity of African populations, leading to many challenges related to the retrieval, representation and findability of such information. To overcome these challenges, we developed the African Genomic Medicine Portal (AGMP), a database that contains metadata on genomic medicine studies conducted on African-ancestry populations. The metadata is curated from two public databases related to genomic medicine, PharmGKB and DisGeNET. The metadata retrieved from these source databases were limited to genomic variants that were associated with disease aetiology or treatment in the context of African-ancestry populations. Over 2000 variants relevant to populations of African ancestry were retrieved. Subsequently, domain experts curated and annotated additional information associated with the studies that reported the variants, including geographical origin, ethnolinguistic group, level of association significance and other relevant study information, such as study design and sample size, where available. The AGMP functions as a dedicated resource through which to access African-specific information on genomics as applied to health research, through querying variants, genes, diseases and drugs. The portal and its corresponding technical documentation, implementation code and content are publicly available.

## 1. Introduction

Modern biomedical research is strongly reliant on data repositories that are well-structured for the storage and retrieval of information. The number of biological and biomedical databases have increased over the last two decades, with resources for diverse organisms, molecules, interactions and pathways. The goal of research is often compounded by the complexity of these different sources and the amount of data that are being gathered from high-throughput technologies, which has also increased in the past two decades. However, low- and middle-income regions, such as Africa, are still not fully capable of tapping into genomic resources for biomedical application due to unmatched resources and needs. There is thus an urgent need to improve facilities, particularly for data generation, processing, storage and access, that will contribute to the implementation of genomic medicine in Africa. Recent initiatives have begun to enhance efforts to overcome the barriers for the use of genomics in Africa, including the Malaria Genomic Epidemiology Network (MalariaGEN) and the Human Heredity and Health in Africa (H3Africa) consortium [1].

Current genomics and biomedical data and metadata in public databases are skewed toward populations of European ancestry. Many African populations remain underrepresented and the data that is available for Africa is not structured in a manner appropriate for the geographical, social and cultural diversity of the continent, with data generally being classified under Sub-Saharan Africa, with little reference to specific populations, or misclassified with other regional groups, as previously described [2,3]. Recent advances in genomics have illustrated the rich genetic diversity of African populations [4,5] and its associated health implications, such as varying lipid profiles [6], predisposition to kidney disease [7] and more [8]. Similar diversity has been observed in pharmacogenomics studies, where, for example, population-specific alleles and frequencies have been found in the genes encoding the cytochrome P450 (CYP) enzymes in Africans, together with structural variants that impact enzymes functionality [9,10,11].

The current underrepresentation of African data and metadata in public databases, as well as the implementation of insufficient classification methods of such data in these public databases, call for the inclusion of more diverse African samples in public genomic reference panels and databases, and highlights the need for a resource that can provide a more granular view of African and African-ancestry populations. This would illustrate the current status of research and implementation on the continent, as well as facilitate continuous progress of genomic medicine in Africa. An accurate representation of actionable genetic variants in Africa is required to facilitate genomic medicine research, including molecular markers that are relevant in the aetiology of diseases, variants that have been associated with specific clinically-supported drug therapies, and variants related to drug response. The ability to query and make use of the currently available data and metadata is critical. Unstructured and heterogeneous data hinders integration efforts and knowledge inference. Moreover, the scattering of data and metadata makes challenging the establishment of a clear genetic determinant map in Africa and affects our capacity to study its genetic diversity and the implication thereof in clinical studies for these population groups at a large scale.

The existing challenges have prompted the development of a fit-for-purpose online database that contains relevant African-specific information related to genomic medicine extracted from public databases and further curated from the literature. This resource will be of interest to the African bioscience community, and genomic medicine stakeholders at large as it may facilitate the implementation of genomic medicine on African-ancestry populations worldwide. The need for an African-specific resource is justified by the extensive genetic diversity among populations across the continent, the data fragmentation across different existing resources, and the lack of appropriate annotation. The proposed resource, the African Genomic Medicine Portal (AGMP) provides curated information to guide the research and application of genomic medicine, pharmacogenomics, and clinical genetics within African-ancestry populations. The AGMP aims to provide a user-friendly resource specific to African research to support genomic medicine in Africa, a centralized root for African studies in genomic medicine available for public use and to supply the community with a well-curated and highly enriched content of genomic medicine that reflects the diversity of the genetic makeup in Africa.

## 2. Methods

### 2.1. Content Mining and Curation

The development of AGMP v1.0 was initiated through a collaborative effort by H3ABioNet, the Pan-African H3Africa Bioinformatics Network members [12]. Two primary data sources were employed for AGMP v1.0, PharmGKB (www.pharmgkb.org, accessed on the 25 July 2019) [13] and DisGeNET (www.disgenet.org, accessed on the 25 July 2019) [14]. The pipeline summarising the different stages for collecting and curating the AGMP data content is presented in Figure 1 and is explained in more detail below. Datasets from these databases were retrieved in plain text format, containing variant-based information on known associations with drug response (PharmGKB) and associations related to disease phenotypes (DisGeNET), as well as the related Pubmed entries (PMID). Gene names were obtained for DisGeNET variants with Ensembl’s Biomart [15] to allow querying of the data by gene symbol. PharmGKB provides gene symbol annotated variants in its raw data files.

To exclude non-African data from AGMP, a custom text-mining pipeline that identifies biomedical literature containing African-related biogeographical and ethnolinguistic terms from Medline abstracts was developed and applied to the retrieved datasets. These terms included a list of 104 biogeographical entities (countries, nationalities, African-ancestry-related population groups such as African-American and Afro-Caribbean) and 146 African ethnolinguistic population labels. The pipeline used the “EDirect” (http://eutils.ncbi.nlm.nih.gov/, accessed on the 10 January 2022) tool to retrieve the PubMed entries in XML format followed by a series of processes to tokenize and Lemmatise the text content. Finally, the list of biogeographical and ethnolinguistic terms was queried and the abstracts that matched at least one of the terms were retained for manual curation. The pipeline is implemented in the Nextflow workflow manager (https://github.com/hothman/text_mining/tree/master/workflow, accessed on the 10 January 2022).

A manual curation step was included in the content curation process due to false positives, which may have resulted from the text-mining pipeline. In addition, Appendix A was extracted from the review of the full-text papers identified in the previous step, including the association significance level (*p*-value), experimental analysis, cohort structure, and more (Appendix A). The manual curation step was also employed to harmonize information related to the biogeographical groups of studies and to extract information about the country of origin of each study. During this process, entries corresponding to studies on populations from African ancestry were harmonized and labeled with one of the following terms: Sub-Saharan Africa, North Africa, West Africa, East Africa, Central Africa, Southern Africa and African-American/Afro-Caribbean. Cohorts of mixed African origins were assigned to Sub-Saharan Africa or Africa depending on the geographical spread of the study samples. Study *p*-values were retrieved either from the PharmGKB metadata field or by recovering it from the main text of the paper.

Drugs derived from the PharmGKB variants–phenotypes list have been annotated by identifying the commercial drug names, the FDA approval status, the indication information, and their conventional names from the International Union of Pure and Applied Chemistry (IUPAC) federation. This information was extracted manually from DRUGBANK [16] (https://go.drugbank.com/, accessed on the 10 January 2022). Drug identifiers from DRUGBANK were retrieved and added to the list of attributes.

All the genes associated with drug response or disease phenotypes were annotated to include information about their gene product, chromosome number and biological function. The gene product was identified by mapping the gene symbol to the corresponding UniProt ID using BioMart. The biological function data were obtained by manual extraction and formatting of the related field from the UniProt database.

Data integration consists of applying the extract, transform and label (ETL) procedure to integrate data from different sources into a relational database. A series of interactive transformation codes were applied for this purpose. Manually curated data were examined for their consistency of using unique terminology in each column. All the data found to describe populations from Non-African ancestry were discarded from the source table. The quality assessment process consists of evaluating the fraction of missing data, the correctness of mapping to the target tables, harmonization of terminologies, harmonization of class labels and the presence of duplicates.

### 2.2. Technical Implementation

The portal can be accessed on the link https://agmp.h3abionet.org (accessed on the 10 January 2022). The search menus have been set up on the basis that most users would need minimum information i.e., the gene name, the variant ID, the disease name or the drug name to easily explore the content of the portal. The portal has been designed with a gene-centric focus which ensures that all the information related to genotype-phenotype data and the associated metadata are attributed to genes at the lowest level of granularity. For example, the user may have access to all the variants belonging to specific gene that include the star-allele types and the SNVs (single-nucleotide variants) related to drug-response or disease phenotype from a single web page on the portal attributed to that gene. With such an implementation, querying, navigating and extracting information from the portal would require only a few prior inputs from the user. Access to data in the portal is facilitated by a combination of an SQLite SQL data storage engine, a Python–Django web framework as an intermediate back-end and an interactive web technology front-end user interface. A Bootstrap framework along with open source JavaScript Libraries and custom CSS are used. The SQLite database consists of six tables linked together using the object relational model (Appendix A).The tables consist of information on research studies and association of human genomic variations, drugs, genes and phenotypes. Data were integrated into the SQLite database with in-house Python scripts that use the Django web framework (https://github.com/h3abionet/african_genomics_medicine_portal, accessed on the 10 January 2022).

## 3. Results

The portal was developed to provide curated information on the genetic variations associated with disease aetiology and treatment in populations of African-ancestry as well as a broad overview of the disease-causing/-associated variants. We utilized PharmGKB and DisGeNET as data sources of the portal mainly for their high-quality content and for providing comprehensive and easy-to-access raw data. DisGeNET includes more than 357,000 variants related to 66,379 bibliography resources. PharmGKB includes 9151 variant-drug associations, including single nucleotides and star alleles described by 3007 papers. An exhaustive search for the African-related data content was difficult to achieve, therefore we opted to automate the identification of African-related biomedical literature to reduce the size of the bibliography resources to be curated. The text mining process allowed us to identify 846 and 164 relevant studies from DisGeNET and PharmGKB respectively. Furthermore, we identified 479 and 2584 variants that are linked to these biomedical literature entries described, respectively, by PhamrGKB and DisGeNET raw data content.

### 3.1. AGMP Data Model and Content

In addition to studies on African populations identified from PharmGKB and DisGeNET, the content of AGMP included data collected from UniProt, DrugBank, Ensembl and PubMed, which was transformed into a relational database containing six tables, as shown on the entity–relationship diagram (Appendix A). The Appendix A explains, in detail, the information provided by each attribute in the relational database. SNVs and star alleles, stored in the ‘Variant’ table, provide information about the variants, and these are associated with phenotypes, which could be related to either the response to a drug or a disease. The star allele variants from PharmGKB were retrieved to account for the particularity of annotating the absorption distribution metabolism excretion (ADME) genes [10]. Note, that a star allele is defined by a set of core variants that define the phenotype associated with the drug response in linkage disequilibrium with other variants, of which presence or absence is not necessarily related to the drug-response phenotype [17,18,19]. The “Variant study” table contains information describing the bio-geographical and the ethnolinguistic attributes for variants in the African population. The other four tables were created to provide additional metadata for the variant–phenotype association. For instance, the “Gene” table stores data about the genes encoding the variants while the “Drug Table” accommodates detailed data for the drugs described by the variant–phenotype associations, including their approval status. Literature resources used by the manual curation process for the retrieval of the additional data are incorporated into he ‘Study’ table. Information about the pharmacokinetic effect on drugs and information about diseases from PharmGKB and DisGeNET respectively are stored in the table ‘Phenotype’.

The manual curation retained 815 studies, including 143 references from PharmGKB and 652 references from DisGeNET that are associated with populations of African ancestry. We identified a total of 556 genes, of which 461 and 95 are related to 552 diseases and 71 drugs, respectively. This represents only a fraction (5.5% and 2%) of the total genes annotated in PhamrGKB and DisGeNET. Many of these diseases are similar but were assigned to different Unified Medical Language System concept unique identifiers in DisGeNET. For instance, “Piebaldism” was described to be associated with SNP rs121913684. “Piebaldism With Sensorineural Deafness” was assigned to another identifier in DisGeNET and was associated with the same rs121913684. The total number of variants associated with the drug response phenotype is 286, of which, 234 correspond to SNVs and 52 are star allele variants.

### 3.2. Data Exploration through the Portal

The frontend of the portal is divided into six main pages (home, about, documentation, search, summary, resources and help), displayed as a menu in the header. The search page has toggle switches that allow a combination of four fields (disease, drug, variant and gene) to be selected, along with the input field to provide search keywords (Figure 2). A complete tutorial on how to use AGMP is included in Appendix A. Access and usage of the data through the AGMP is available under the Attribution-NonCommercial-ShareAlike 4.0 International License (CC BY-NC-SA 4.0), as both PharmGKB and DisGeNET content are distributed for public use under the same type of license. AGMP seeks to comply with FAIR principles in its development cycle. To achieve this, the portal includes documentation, public code sharing, data and code archiving and licensing. Formal FAIRification from the data content and web accessibility point of view will be addressed further in upcoming versions of the portal.

### 3.3. Representation of African Data in AGMP

Studies about genetic markers from populations of African ancestry have been conducted by different research teams from around the world. The ‘country’ data of AGMP aims to assign the annotated variants in the database to a geographical entity. A geographical entity is an indicator of where the study participants were recruited, including names of the country and the region. The annotation is dependent on the level of detail provided by a study, usually in the publication. Many of these studies have used a mixed population either of total African individuals or combined with non-Africans. For example, Zimmerman et al. [20] studied the polymorphism of chemokine receptor 5 (CCR5) in a group of mixed ethnicity that included Africans. However, they did not provide details about the exact origin of the African cohort, therefore their geographical entity was assigned to “Africa”. For others, more fine-grained geographical locations were provided.

AGMP includes data derived from 37 African countries, though we noticed a significant imbalance of identified variants number between the different regions of Africa (Figure 3A). With 66% of all the annotated variants, countries from North Africa are, by far, the most represented in the dataset. We also noticed an important gap between the central African region and the rest of the continent, with only 52 variants corresponding to just 2.4%, derived from countries in this region. This is the least represented region in the AGMP dataset despite its demographic and historical importance in Africa. Tunisia has the most annotated variants, with 568 variants, representing 23% of the entire dataset (Figure 3B).

### 3.4. Pharmacogenetic-Related Phenotypes in AGMP

Variants identified in PharmGKB are associated with drug response phenotypes for 50 drugs approved for specific treatments (Figure 4). Warfarin is the drug with the highest number of associated variants with a total of 192, representing about 7% of the entire dataset. Anticoagulant drugs are the most represented therapeutic class associated with 37% of the total annotated variants.

## 4. Discussion

The first version of the AGMP was implemented to provide high-quality and reliable information on genetic variants from African populations and populations of African ancestry for users with interests in genomic medicine. The primary sources of data were PharmGKB and DisGeNET supplemented with manual curation from the literature and links to other databases. These provide rich metadata for genetic variants in a standard format and use controlled vocabularies. The two databases are also recognized for the high quality of their content because of their rigorous processes of curation. For example, the PharmGKB curation panel of experts regularly verify the biomedical literature to integrate novel variants or update the information about the association with drug-response phenotypes [21], while DisGeNET aggregates data from eight databases including UniProt, ClinVar, GWAS catalog and GWASdb [22].

Data profiling and curation for AGMP have highlighted some drawbacks in the current conception of DisGeNET and PharmGKB, as well as the practices of describing the experimental protocol and presenting data in genetic and genomic studies. DisGeNET was intended to give a broad overview of variant-phenotype associations. The data, however, are presented as a binary association without taking into consideration the ethnic composition of the original study participants, hence our need to use text analysis and manual curation to extract data and metadata. In contrast, PharmGKB has adopted specific guidelines to assign a biogeographical group to each association [23]. According to this scheme, Africa is represented by “Sub-Saharan African” and ”African American/Afro-Caribbean” groups, while the North African region was assigned to the “Near eastern” group. These annotation guidelines are inadequate, as the Sub-Saharan African region is not ethnically and culturally homogenous [5]. The labels also ignore the admixture landscape of the North African population [24,25,26] that is significantly different from the broad near-eastern population. In addition to these issues, manual curation showed that the description of biogeographical information is either lacking, inconsistent or inaccurate in the biomedical literature. Providing a standardized ethnolinguistic ontology will be beneficial in annotating genomic medicine data. Some of the variants in AGMP were assigned to the Sub-Saharan and North African geographical regions because we could not find enough information to annotate the country of origin. Since it is difficult to use a global annotation scheme for assigning biogeographical data to the genotype–phenotype association, it would be valuable to establish clear guidelines on how to ensure broad coverage of the genetic diversity in Africa. This highlights the need to integrate geographical information system data with genomic data, which could and would provide a valuable asset to be linked to the AGMP portal.

The under representation of some African regions in public databases has been extensively discussed in several papers [2,3,27,28]. Very few entries have been assigned to the central-African region compared with the other African regions. It is not clear, however if this unbalance is caused by the lack of reliable studies originating in central Africa or the lack of data related to the region generated by consortia or submitted to databases. Moreover, to facilitate automated extraction of genotype-phenotype associations, collection and reporting of standardized phenotypic data was also highlighted as being a challenge.

The disproportionate skew in the geographical distribution of data may be due to numerous factors; one of the main factors being the data representation in PharmGKB and DisGeNET. Both resources contain more data from North Africa as a result of many mixed studies with southeastern European, Middle Eastern and Francophone countries. In addition, DisGeNET contains a large proportion of case studies from North Africa and has less representation of other regions of Africa. This skew does not exist in all other biomedical databases (e.g., GWAS Catalog), and as more resources are incorporated into the Portal, the distribution should become more evenly spread. As such, the Portal will be continually updated and new data sources will be included in the future. One of the main aims of developing the Portal was to highlight such knowledge gaps and discrepancies within Africa so that we can better address these research questions in the future.

We expect the AGMP to be a crucial source of information for the ongoing African research activities in pharmacogenetics and genomic medicine. As a product of H3ABioNet, a member of the H3Africa consortium, we have the opportunity to promote the AGMP among a large community of African genetics researchers, which will allow us to update, improve and diffuse its content. Besides its informative added value, throughout the processes of data profiling, curation and web application development, the project enabled H3ABioNet to develop a solid core of experts, including bioinformaticians, curators, computer scientists, clinical biologists and data scientists. The core group will have a critical role in ensuring the sustainability and update of the portal and to maintain its relevance to genomic medicine-related activities in Africa.

## 5. Conclusions

The AGMP is a user-friendly resource that allows users to query curated African genomic variants linked to either clinical phenotypes or to drug response using gene, disease, variant and drug names. This portal should be of interest to the African scientific community, physicians and genomic medicine stakeholders worldwide, as it provides informative data for those working with individuals of African descent. In addition, our work emphasizes the importance of population sampling coupled to GIS data acquisition that can improve the granularity of the geographical location to better represent the genetic diversity in Africa. It would also help in linking non-genetic factor contributions to phenotype expression.

Two strategies have been proposed to extend the capacity of AGMP and improve the content and features of the resource. In the first, a curation group is established to run updated versions of the data mining, cleaning and integration workflows with more robust methods. This includes the training of predictive models that could process, in high-throughput mode, Medline entries that are not necessarily described in DisGeNET or PharmGKB. This will also allow communicating information about variants that are specific to Africa or with higher prevalence in Africa. Additionally, the group will implement reliable methods to automate the processes of data extraction and curation by utilizing APIs and entity–relationship detection from the biomedical literature. The data curation team will also determine the feasibility-to-adopt in integrating new data sources to capture African data in the GWAS catalog and GWASdb. Second, we aim to increase engagement with the scientific community involved in genomic medicine research in Africa by promoting the portal to the wider community, establishing an extended network of experts and curators, and adapting the portal user interface to allow for direct submission of the data. Finally, in order to provide more detail related to the studies captured in the Portal, a notes section will be included in upcoming versions of the Portal, providing additional study information such as study type and sample size.

The portal’s primary aim is to facilitate genomic medicine research by researchers and clinicians in Africa or those working on African-ancestry populations. Secondarily, the portal may also serve as a guide for healthcare workers for diagnosis and prognosis of diseases and adverse drug reactions within their patient populations in the future and improve their understanding of the genetic underpinnings of precision medicine in Africa. However, it should be noted that the portal is a research tool and should not be used for clinical decision-making; rather, it is intended to provide doctors with the developing knowledge of pharmacogenetics and precision medicine that may be relevant for their patient population.

## Figures and Tables

**Figure 1 jpm-12-00265-f001:**
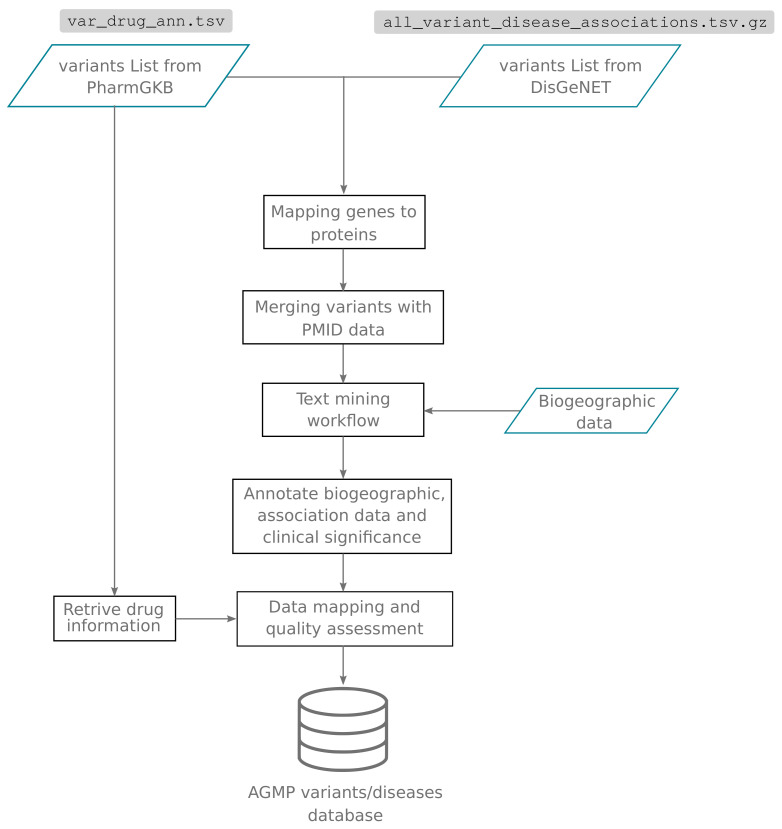
AGMP v1.0 pipeline for data mining, curation and cleaning.

**Figure 2 jpm-12-00265-f002:**
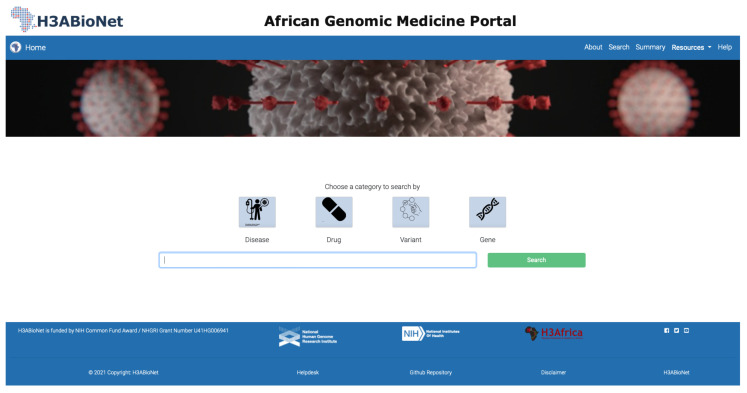
Screenshot of the search menu of the AGMP. Users can explore the data by querying diseases, drugs, variants and genes.

**Figure 3 jpm-12-00265-f003:**
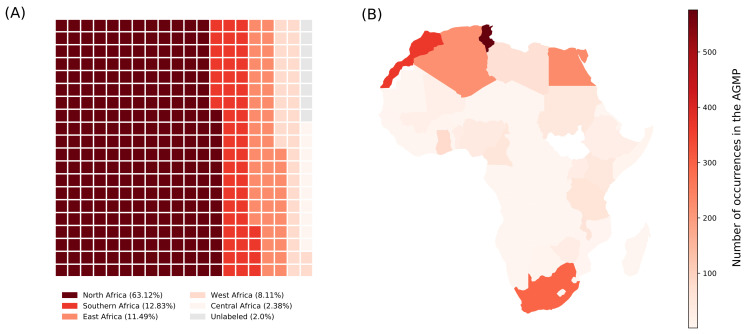
Representation of African countries in AGMP. (**A**) The waffle plot shows the ratio of occurrence in the geographical location annotation data by African regions (North Africa, Southern Africa, East Africa, West Africa and Central Africa). The assignment of countries at regional representation was established according to the UN classification. The squares are arbitrary units of proportion. (**B**) Representation of countries according to the number of corresponding variants.

**Figure 4 jpm-12-00265-f004:**
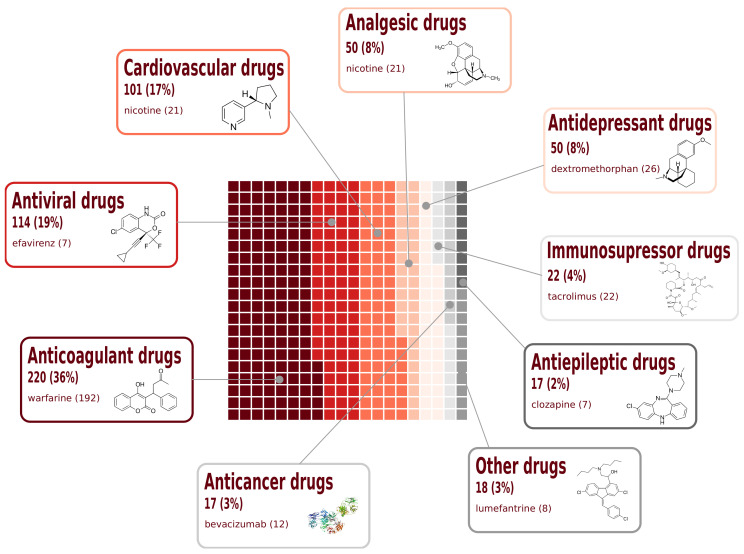
Representation of therapeutic drug classes in AGMP. The total number of variant-drug response phenotype associations is given in the annotation boxes. The shading color levels represent the different classes to visually show the number of variants in the database.

## Data Availability

Access to the AGM portal is available (https://agmp.h3abionet.org, accessed on 10 January 2022). The code developed for the web implementation can be found: https://github.com/h3abionet/african_genomics_medicine_portal, accessed on 10 January 2022. The list of biogeographical and ethnolinguistic terms used to filter Medline abstracts are available from this link: https://raw.githubusercontent.com/hothman/text_mining/master/data/query_terms.txt, accessed on 10 January 2022. The workflow used for mining the biomedical Medline abstract can be found: https://github.com/hothman/text_mining/tree/master/workflow, accessed on 10 January 2022.

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
