# Peer review of "African Genomic Medicine Portal: A Web Portal for Biomedical Applications"

_jpm, 2022, doi:10.3390/jpm12020265_

Round 1

Reviewer 1 Report

Othman and colleagues have used two publically available databases, PharmGKB and DisGeNET, to extract metagenomic data related to African populations to establish the African Genomic Medicine Portal (AGMP), which contains these data in a curated form. The effort is noteworthy, and access to genetic and medical data from African populations is highly useful and timely. In the manuscript, the authors have outlined the AGMP database clearly. Reading through the manuscript, I have two comments:

1. A perceived discrepancy between the authors' effort to establish a comprehensive genomic, medical and pharmacological database for all populations on the continent, with the majority of this data coming from very limited geographic regions. For example, if I read the manuscript correctly, the combined data from West and Central Africa are represented by only 11% of the total data, with what seems to be 63% of all data coming from one single region, Tunisia.  While, the authors themselves cannot compensate for this discrepancy, I still would like to see more discussion about it and its potential impact on data interpretability. 

2. Conceptaully, it would have been beneficial to have summary of the results presented in the databases, rather than a p-values and PMID # to the abstract. However, this is somewhat a minor comment. 

Author Response

1. A perceived discrepancy between the authors' effort to establish a comprehensive genomic, medical and pharmacological database for all populations on the continent, with the majority of this data coming from very limited geographic regions. For example, if I read the manuscript correctly, the combined data from West and Central Africa are represented by only 11% of the total data, with what seems to be 63% of all data coming from one single region, Tunisia. While, the authors themselves cannot compensate for this discrepancy, I still would like to see more discussion about it and its potential impact on data interpretability.

Reply

Thank you for this comment, the reviewer highlights an important point, and we have added a paragraph in the discussion to further discuss this. The distribution is likely due to the data captured in the original resources, however, it should become more evenly spread as more resources are added. In addition, it is important to note that the aim of the Portal is to highlight such data and knowledge discrepancies as highlighted by the reviewer, in order to address these gaps in future research.

The paragraph below was added to the discussion page (9), lines (290-301).

The disproportionate skew in the geographical distribution of data may be due to numerous factors, one of the main factors being the data representation in PharmGKB and DISGENET. Both resources contain more data from North Africa as a result of many mixed studies with southeastern European, Middle Eastern, and Francophone countries. In addition, DISGENET contains a large proportion of case studies from North Africa and has less representation of other regions of Africa. This skew does not exist in all other biomedical databases (e.g. GWAS Catalog), and as more resources are incorporated into the Portal, the distribution should become more evenly spread. As such, the Portal will be continually updated and new data sources will be included in the future. One of the main aims of developing the Portal was to highlight such knowledge gaps and discrepancies within Africa so that we can better address these research questions in the future.

2. Conceptually, it would have been beneficial to have summary of the results presented in the databases, rather than a p-values and PMID # to the abstract. However, this is somewhat a minor comment.

Reply

To simplify data curation and present data in an easily digestible format, summaries were excluded from the results. The PMID links that redirect the user to an abstract serve this purpose. However, the upcoming version of the Portal will also contain a notes section in the results, which will contain additional details of each study/result, including details such as sample size, study type etc.

The following paragraph was added to the conclusion section page (10) lines, (334-336).

In order to provide more detail related to the studies captured in the Portal, a notes section will be included in upcoming versions of the Portal, providing additional study information such as study type and sample size.”

Reviewer 2 Report

The authors of the manuscript set themselves the goal of i.a. promotion the portal to the wider community.
I would suggest showing a concrete example how to use the portal described in the manuscript - for example show a step-by-step diagram of using the portal (you can show such a figure in the "supplementary file"). I think it will facilitate the use of the portal by doctors and specialists who have not used bioinformatics databases so far. 

I would suggest that in the chapter "conclusion" add one two sentences about the clinical application of the presented portal. 

Author Response

The authors of the manuscript set themselves the goal of i.a. promotion the portal to the wider community. I would suggest showing a concrete example how to use the portal described in the manuscript - for example show a step-by-step diagram of using the portal (you can show such a figure in the "supplementary file"). I think it will facilitate the use of the portal by doctors and specialists who have not used bioinformatics databases so far.

Reply

We agree with the reviewer’s critique, and have thus added a tutorial document as a supplementary file to the manuscript. We also aim to develop a tutorial video in the future which will be hosted on the portal website, along with the newly attached tutorial, which can also be accessed via the website.

I would suggest that in the chapter "conclusion" add one or two sentences about the clinical application of the presented portal.

Reply

As requested, the authors have expanded on the topic in the conclusion page (10), lines (342-350).

The portal’s primary aim is to facilitate genomic medicine research by researchers and clinicians in Africa or those working on African-ancestry populations. Secondarily, the portal may also serve as a guide for healthcare workers for diagnosis and prognosis of diseases and adverse drug reactions within their patient populations in the future and improve their understanding of the genetic underpinnings of precision medicine in Africa. However, it should be noted that the portal is a research tool and should not be used for clinical decision-making, rather, it is intended to provide doctors with the developing knowledge of pharmacogenetics and precision medicine that may be relevant for their patient population.”

Reviewer 3 Report

Paper by Othman et al. publicize a new web resource devoted to African Genomic Medicine Portal (AGMP), a database that contains metadata on genomic medicine studies conducted on African-ancestry populations. To develop this data source Authors used two established database  (PharmGKB and DisGeNET) to retrieve data on genomic variants associated with disease aetiology or treatment in the context of African-ancestry populations. 

Data reported in the paper appear convincing and implementation and use of this new tool will promise to increase knowledge on genetic background of African and African-ancestry populations in a view of an approach of a tailored medicine offered worldwide. In my opinion research described in this paper is well conceived and developed.

Author Response

No revisions were requested by reviewer 3

Reviewer 4 Report

In this article, authors address an important achievement to the incoming genomic studies in the African population: the elaboration of a public and user-friendly correlational database for genomic, phenotypic, and pharmacological data based in African-population studies. This resource aims to be useful in health research in a large population still to be an important player in population genomic studies.

Overall, the article is quite easy to read and understand. The open access code and frontend created for this tool are remarkable. I consider is a great step ongoing that can be improved in the future when large genomic studies settle an increased number of sequenced samples for the African population, so this database improves drastically without the existing major problem that suppose the poor information for the general African population in DisGeNet and PharmaGKB. Also, some automation should be addressed for future updates of the database, as detailed by the authors, this database has a lot of manual curation and manual input (such DrugBank annotations), and this may be a concern when externals datasets are upgraded. Some comment regarding this issue should be attached in a line or two in the text as future proof methodologies.

Some little details in the text are marked by line number.

Line 182 – there is a dot outlier in the text.

Line 184 – ADME should need a citation.

Line 194 – As three of the four metadata tables have been described, it could be useful to also add Phenotype table with a brief description.

Line 286 – remove extra “Moreover,”

Figure 4 colouring is a bit difficult to understand. Perhaps it could be easier if the names of each group of drugs or the marker around each group matches the same colouring for that group of drugs in the squares representations, i.e Antiepileptic drugs in darker grey; other drugs in a lighter grey, and so on.

Author Response

Also, some automation should be addressed for future updates of the database, as detailed by the authors, this database has a lot of manual curation and manual input (such DrugBank annotations), and this may be a concern when externals datasets are upgraded. Some comment regarding this issue should be attached in a line or two in the text as future proof methodologies.

We have added a phrase regarding this issue in lines 330-332 (Conclusion section)

Line 182 – there is a dot outlier in the text.

This is now fixed in the revised version

Line 184 – ADME should need a citation.

We added a reference by da Rocha et al (2021)

Line 194 – As three of the four metadata tables have been described, it could be useful to also add Phenotype table with a brief description.

The phenotype table is described briefly (lines 194-196)

Line 286 – remove extra “Moreover,”

This is now fixed in the revised version

Figure 4 colouring is a bit difficult to understand. Perhaps it could be easier if the names of each group of drugs or the marker around each group matches the same colouring for that group of drugs in the squares representations, i.e Antiepileptic drugs in darker grey; other drugs in a lighter grey, and so on.

This is now modified in the revised version

Round 2

Reviewer 1 Report

The authors have responded to my critiques. I do not have any further comments.